# Measurement-Device-Independent Quantum Key Distribution Based on Decoherence-Free Subspaces with Logical Bell State Analyzer

**DOI:** 10.3390/e25060869

**Published:** 2023-05-29

**Authors:** Jun-Hao Wei, Xin-Yu Xu, Shu-Ming Hu, Qing Zhou, Li Li, Nai-Le Liu, Kai Chen

**Affiliations:** 1Hefei National Research Center for Physical Sciences at the Microscale and School of Physical Sciences, University of Science and Technology of China, Hefei 230026, China; weijunhao@mail.ustc.edu.cn (J.-H.W.);; 2CAS Center for Excellence in Quantum Information and Quantum Physics, University of Science and Technology of China, Hefei 230026, China; 3Hefei National Laboratory, University of Science and Technology of China, Hefei 230088, China

**Keywords:** quantum key distribution, measurement-device-independent, decoherence-free subspace, logical Bell state

## Abstract

Measurement-device-independent quantum key distribution (MDI-QKD) enables two legitimate users to generate shared information-theoretic secure keys with immunity to all detector side attacks. However, the original proposal using polarization encoding is sensitive to polarization rotations stemming from birefringence in fibers or misalignment. To overcome this problem, here we propose a robust QKD protocol without detector vulnerabilities based on decoherence-free subspaces using polarization-entangled photon pairs. A logical Bell state analyzer is designed specifically for such encoding. The protocol exploits common parametric down-conversion sources, for which we develop a MDI-decoy-state method, and requires neither complex measurements nor a shared reference frame. We have analyzed the practical security in detail and presented a numerical simulation under various parameter regimes, showing the feasibility of the logical Bell state analyzer along with the potential that double communication distance can be achieved without a shared reference frame.

## 1. Introduction

Quantum key distribution (QKD) [1,2,3], as one of the most outstanding and mature applications of quantum information science, allows for the establishment of shared secure keys among two remote communication parties. Since the proposal of the first QKD protocol—BB84 protocol [4], enormous amount of endeavors have been dedicated to increase the performance of QKD. So far, the communication distance has reached 1000 km [5] and the bit rate has exceeded 110 Mb$\cdot s^{-1}$ [6].

Despite the unconditional security guaranteed by postulates of quantum mechanics in theory, the practical security of QKD can be compromised due to the deviations between real-life implementations and idealized models in security proofs. As photon detectors are usually complicated and cumbersome to calibrate, they are vulnerable to many kinds of sophisticated attacks [7,8,9]. Fortunately, measurement-device-independent QKD (MDI-QKD) protocol [10] closes all detector side channels and has attracted numerous studies in various scenarios due to its brevity. The original MDI-QKD protocol uses polarization encoding [11,12,13]; moreover, many other encoding forms have been well-studied both theoretically and experimentally [14,15,16,17,18].

In practice, one of the most significant problems that hinders the development of polarization-encoding QKD systems is polarization rotation stemming from birefringence effects in single mode fibers [19], as there exist external perturbations and small fluctuations in their material anisotropy [20]. The scheme proposed by Boileau et al. [21], which is a variant of BB84 protocol, aimed to overcome this problem by encoding two timely separated photons into the so-called decoherence-free subspace (DFS), span$\{|0_{L}\rangle =\left|HV\rangle ,\right|1_{L}\rangle =|VH\rangle \}$, where the subindex *L* stands for logical, and *H* and *V* denote horizontal and vertical polarized photons, respectively. For the scheme to work, the delay caused by polarization mode dispersion is supposed to be smaller than the coherence time of the photon, which holds in most practical circumstances currently, so the birefringence acts just as a rotation U(t) on the polarization modes. Another constraint is that the photon pair experiences collective rotation $U{\left(t\right)}^{⊗2}$, thus the orthogonality between horizontal and vertical polarization mode is conserved. We remark that this constraint is significantly weaker than that of reference-frame-independent QKD (RFI-QKD) protocol [22], which is only insensitive to rather slow variance of the reference frame during the whole communication process. In a recent work by Tang et al. [23], the divide-and-conquer strategy is applied to develop the free-running RFI-QKD scheme, which weakens the aforementioned constraint on drifting speed to some extent, but its tolerance of reference frame drifting is still not as strong as Boileau et al.’s scheme.

In this work, we generalize the scheme of Boileau et al. to an MDI version by solving two main obstacles: lack of logical Bell state measurement and decoy-state analysis for polarization-entangled photon pairs. We develop a logical Bell state analyzer and MDI-decoy-state method [19,24,25] specifically for practical parametric down-conversion (PDC) sources. Our logical Bell state analyzer is easy to implement with current technology and exploits cross-Kerr nonlinearities, which have found wide applications in quantum information processing tasks [26,27,28,29]. In addition, the MDI-decoy-state method enables us to estimate the contribution from single-photon-pair states more accurately [30,31,32], greatly extending the communication distance. Our protocol enjoys MDI characteristics and RFI features simultaneously, with built-in robustness against collective polarization rotation noise. The scheme does not require complex entanglement measurements and is stable even in presence of some imperfections of the logical Bell state analyzer.

The rest of this paper is organized as follows. We introduce the MDI-QKD protocol based on DFS and investigate its robustness in Section 2. In Section 3, we present the setup of the logical Bell state analyzer. In Section 4, we perform a security analysis on the MDI-QKD system, followed by simulation results of the key rate in Section 5. In Section 6, we discuss several experimentally relevant problems and potential improvements of the protocol. Finally, we conclude the paper in Section 7.

## 2. Protocol

In the scheme of MDI-QKD protocol based on DFS, two communication parties named Alice and Bob generate polarization-entangled photon pairs independently. The key information is encoded into the relative phase between $|0_{L}\rangle$ and $|1_{L}\rangle$. After some pre-processing operations, they send the states to an untrusted relay Charlie that is located in the middle. Several post-processing operations followed by logical Bell state measurement (BSM) are expected to be performed by the measurement site, as illustrated in Figure 1. Even in presence of collective rotation induced by birefringence or misaligned reference frame, Alice and Bob’s key information can be correctly correlated conditioned on the measurement results announced by the relay. Detailed steps of our protocol with decoy-state method are presented below.
**State preparation**: In each round, Alice pumps her phase randomized type-II PDC source using appropriate intensities to generate polarization-entangled states with half of the expected photon pair number generated by one pulse, $\lambda_{a}$, randomly selected from {μ,ν,0} according to probability distribution $\left\{p_{\mu },p_{\nu },p_{0}\right\}$. The single-photon-pair state prepared by Alice is $\frac{1}{\sqrt{2}}\left({|HV\rangle }_{12}+\exp\left[i(\pi K_{a}+\frac{\pi }{2}B_{a})\right]{|VH\rangle }_{12}\right)$, where $K_{a}\in \left\{0,1\right\}$ is the encoded bit, $B_{a}\in \left\{0,1\right\}$ corresponds to bases {Z,X}, 1 and 2 label the two optical modes of the PDC source, and mode 2 is delayed by Δt with respect to mode 1. Then, she delays her vertical polarization mode by $\Delta t^{\prime }$, where $\Delta t^{\prime }<\Delta t$. Similarly, the single-photon-pair state prepared by Bob is $\frac{1}{\sqrt{2}}\left({|HV\rangle }_{34}+\exp\left[i(\pi K_{b}+\frac{\pi }{2}B_{b})\right]{|VH\rangle }_{34}\right)$ where mode 4 is also delayed by Δt with respect to mode 3, and $\lambda_{b}$ is randomly chosen from {μ,ν,0} with the same probability distribution. Then, he delays his vertical polarization mode by $\Delta t^{\prime }$ as well.**Measurement**: Alice and Bob send the signals to Charlie, who is supposed to perform
(1)Polarization randomization using a set of wave plates to make the protocol independent of specific environment and reference frame,(2)Delay for the horizontal polarization mode of both incoming signals by $\Delta t^{\prime }$, such that both photons of the states in DFS are delayed once,(3)Phase scrambling to project the single-photon-pair states into the DFS, which can be done by exploiting Pockel cells driven by quantum random number generators (QRNG),(4)Logical BSM using the logical Bell state analyzer.Additionally, Charlie needs to set the polarization controllers in the logical BSM apparatus to act as nothing in some randomly selected rounds retained for parameter estimation, which we call sampling rounds. The other rounds where both polarization controllers act as half-wave plates are named BSM rounds. Charlie publicly announces the results of parity check measurements and click patterns of the four single-photon-detectors in all rounds, together with the time-bins in which the detectors click and the location of sampling rounds.**Postselection**: Alice and Bob postselect the BSM rounds where one two-fold coincidence detection is followed by another two-fold coincidence detection after Δt. They determine which logical Bell states are the input states successfully projected onto according to Charlie’s announcement (see Section 3). Events with unsuccessful logical BSM are discarded.**Sifting**: The parties announce $\lambda_{a,b}$ and $B_{a,b}$ in the remaining rounds via an authenticated public channel. After discarding the rounds where their bases are unmatched, one communication party, say, Bob, should flip part of his bits to make his bit strings correctly correlated with Alice’s, depending on the logical Bell states identified (see Table 1).**Parameter estimation**: For BSM rounds, Alice and Bob estimate the quantum bit error rate $E_{\mu \mu }^{Z}$ in *Z* basis with intensity setting $\left(\lambda_{a},\lambda_{b}\right)=\left(\mu ,\mu \right)$. Raw data in *Z* and *X* basis are used to estimate the single-photon-pair yield $Y_{11}^{Z}$ and single-photon-pair QBER $e_{11}^{X}$, respectively, via decoy-state method introduced in Section 4. For sampling rounds, they use data with intensity setting $\left(\mu ,0\right)$ and $\left(0,\mu \right)$ to estimate the probability that their single-photon-pair states are projected into the DFS, denoted as $p_{a}$ and $p_{b}$, respectively.**Key distillation**: The parties use data of BSM rounds in *Z* basis with intensity setting $\left(\mu ,\mu \right)$ to generate a key. They run error correction and privacy amplification based on $p_{a},p_{b},E_{\mu \mu }^{Z},Y_{11}^{Z}$ and $e_{11}^{X}$ to distill the final secure key.

We now show the correctness and robustness against collective polarization rotations of our protocol. Since the birefringence effect experienced by Alice and Bob’s states are independent, here we focus on the evolution of Alice’s states in the DFS, which we write as $\alpha |HV_{T}\rangle +\beta |V_{T}H\rangle$ without loss of generality, where ${\left|\alpha \right|}^{2}+{\left|\beta \right|}^{2}=1$. The subscript *T* indicates the delay by $\Delta t^{\prime }$. As the above state transmitting through fibers, the polarization modes experience a rotation represented by an element in the group U(2). The resulting polarizations can be expanded in Charlie’s own basis $\left\{|H^{\prime }\rangle ,|V^{\prime }\rangle \right\}$ as following up to a global phase,
(1)$$\begin{array}{cc} \; & U|H\rangle =e^{-i\left(\phi +\omega \right)/2}\cos\frac{\theta }{2}|H^{\prime }\rangle +e^{i\left(\phi -\omega \right)/2}\sin\frac{\theta }{2}|V^{\prime }\rangle ,\\ \; & U|V\rangle =-e^{-i\left(\phi -\omega \right)/2}\sin\frac{\theta }{2}|H^{\prime }\rangle +e^{i\left(\phi +\omega \right)/2}\cos\frac{\theta }{2}|V^{\prime }\rangle , \end{array}$$
where $\theta \in \left[0,\pi \right],\phi \in \left[0,2\pi \right),\omega \in \left[0,4\pi \right)$. Then, Charlie’s delay operation for his horizontal polarization mode $H^{\prime }$ would lead the states to evolve as
(2)$$\begin{array}{cc} \; & U^{⊗2}\left(\alpha |HV_{T}\rangle +\beta |V_{T}H\rangle \right)\overset{\mathrm{Delay}\;H^{\prime }}{⟶}\\ \; & \alpha \left(\cos^{2}\frac{\theta }{2}|H_{T}^{\prime }V_{T}^{\prime }\rangle +\sin\frac{\theta }{2}\cos\frac{\theta }{2}e^{i\phi }|V^{\prime }V_{T}^{\prime }\rangle -\sin\frac{\theta }{2}\cos\frac{\theta }{2}e^{-i\phi }|H_{T}^{\prime }H_{TT}^{\prime }\rangle -\sin^{2}\frac{\theta }{2}|V^{\prime }H_{TT}^{\prime }\rangle \right)\\ + & \beta \left(\cos^{2}\frac{\theta }{2}|V_{T}^{\prime }H_{T}^{\prime }\rangle +\sin\frac{\theta }{2}\cos\frac{\theta }{2}e^{i\phi }|V_{T}^{\prime }V^{\prime }\rangle -\sin\frac{\theta }{2}\cos\frac{\theta }{2}e^{-i\phi }|H_{TT}^{\prime }H_{T}^{\prime }\rangle -\sin^{2}\frac{\theta }{2}|H_{TT}^{\prime }V^{\prime }\rangle \right)\\ \equiv & \left[\left(\delta_{1}+1\right)/2\right]\left(\alpha |H_{T}^{\prime }V_{T}^{\prime }\rangle \right.\left.+\beta |V_{T}^{\prime }H_{T}^{\prime }\rangle \right)+\left[\left(\delta_{1}-1\right)/2\right]\left(\alpha |V^{\prime }H_{TT}^{\prime }\rangle +\beta |H_{TT}^{\prime }V^{\prime }\rangle \right)\\ + & \left[\left(\delta_{2}+\delta_{3}\right)/2\right]\left(\alpha |H_{T}^{\prime }H_{TT}^{\prime }\rangle +\beta |H_{TT}^{\prime }H_{T}^{\prime }\rangle \right)+\left[\left(\delta_{2}-\delta_{3}\right)/2\right]\left(\alpha |V^{\prime }V_{T}^{\prime }\rangle +\beta |V_{T}^{\prime }V^{\prime }\rangle \right) \end{array}$$
where $\delta_{1}=\cos\theta$, $\delta_{2}=i\sin\theta \sin\phi$, and $\delta_{3}=-\sin\theta \cos\phi$ [21]. The parameters satisfy ${\left|\delta_{1}\right|}^{2}+{\left|\delta_{2}\right|}^{2}+{\left|\delta_{3}\right|}^{2}=1$ and are completely dependent on the explicit form of *U*. From Equation (Equation 2), we can see that Charlie can recover the encoded key information with certainty by postselecting the states where both photons are delayed once, i.e., they are separated by Δt. In addition, in order to guarantee successful postselection, Charlie needs to perform polarization randomization to both photons before his delay operations, otherwise $\left(\delta_{1}+1\right)/2$ may vanish in all rounds in some extreme circumstances. As the rotation is equivalent to be uniformly distributed over U(2) after polarization randomization, we can calculate the probability of postselection by means of the Harr measure,
(3)$$\begin{array}{c} \frac{1}{16\pi^{2}}\int_{0}^{2\pi }d\phi \int_{0}^{4\pi }d\omega \int_{0}^{\pi }{\left(\cos^{2}\frac{\theta }{2}\right)}^{2}\sin\theta d\theta =\frac{1}{3}. \end{array}$$Likewise, Charlie has the probability of 1/3 to postselect Bob’s encoded states. Therefore, the joint probability of successful postselection of both parties’ states is 1/9.

The RFI characteristic of our protocol is embodied in the fact that Alice, Bob, and Charlie’s operations are all performed in their local reference frame. Combined with the logical Bell state measurement apparatus introduced in the next section, our protocol is thus similar to the original MDI-QKD protocol and can overcome the problem of polarization rotations induced by birefringence effects or reference frame misalignment.

## 3. Logical Bell State Measurement

The crucial component of MDI-QKD protocol is the projection of Alice and Bob’s input signals onto maximally entangled states. Thus, in the protocol based on 2-dimensional DFS spanned by $\{|0_{L}\rangle =\left|HV\rangle ,\right|1_{L}\rangle =|VH\rangle \}$, one has to perform logical BSM. We have designed a logical Bell state analyzer, which is shown in Figure 2. The apparatus is able to completely discriminate four logical Bell states
(4)$$\begin{array}{c} |\Phi_{L}^{\pm }\rangle =\frac{1}{\sqrt{2}}\left(|HVHV\rangle \pm |VHVH\rangle \right), \end{array}$$
(5)$$\begin{array}{c} |\Psi_{L}^{\pm }\rangle =\frac{1}{\sqrt{2}}\left(|HVVH\rangle \pm |VHHV\rangle \right), \end{array}$$
each of which contains two logical qubits encoded by two orthogonally polarized photons.

In Figure 2, we represent the optical modes by orange wave packets, and the two probe coherent pulses should overlap with the time-bins, respectively. The vertical lines together with circles denote cross-phase modulation via Kerr effect. The $D_{1H},D_{1V},D_{2H},and\;D_{2V}$ denote four single-photon detectors. We restrict our discussion to single-photon-pair states in this section. Due to postselection, one only needs to consider the incoming states from either party of the form $\frac{1}{\sqrt{2}}\left(|HV\rangle +e^{i\phi }|VH\rangle \right),\phi \in \left\{0,\pi /2,\pi ,3\pi /2\right\}$. Remember that photons 1 and 3 are in the same time-bin; photons 2 and 4 are in another time-bin delayed by Δt. The key observation is that photons 1 and 3 have identical polarization, i.e., even parity, in $|\Phi_{L}^{\pm }\rangle$, whereas they have orthogonal polarization, i.e., odd parity, in $|\Psi_{L}^{\pm }\rangle$; the same is true for photons 2 and 4. Therefore, one can divide the four logical Bell states into two groups with a parity check measurement (PCM).

The PCM exploits two probe coherent states, each of which is aligned with one time-bin, along with cross-Kerr nonlinearities. As shown in Figure 2, the phase of the probe states are modulated by the photon numbers in the corresponding paths via Kerr effect. Without loss of generality, we suppose the polarization state of photons 1 and 3 are ${a|H\rangle }_{1}{|H\rangle }_{3}+{b|H\rangle }_{1}{|V\rangle }_{3}+{c|V\rangle }_{1}{|H\rangle }_{3}+{d|V\rangle }_{1}{|V\rangle }_{3}$, where ${\left|a\right|}^{2}+{\left|b\right|}^{2}+{\left|c\right|}^{2}+{\left|d\right|}^{2}=1$. Then the evolution of it together with the upper coherent state in Figure 2 is as follows [33]:(6)$$\begin{array}{cc} \; & \left({a|H\rangle }_{1}{|H\rangle }_{3}+{b|H\rangle }_{1}{|V\rangle }_{3}+{c|V\rangle }_{1}{|H\rangle }_{3}+{d|V\rangle }_{1}{|V\rangle }_{3}\right)|\alpha \rangle \\ ⟶ & \left({a|H\rangle }_{1}{|H\rangle }_{3}+{d|V\rangle }_{1}{|V\rangle }_{3}\right)|\alpha \rangle +{b|H\rangle }_{1}{|V\rangle }_{3}|\alpha e^{-2i\theta }{\rangle +c|V\rangle }_{1}{|H\rangle }_{3}|\alpha e^{+2i\theta }\rangle . \end{array}$$For homodyne measurement of the *X* quadrature, the phase shift +2θ and −2θ will produce identical probability distribution of the outcome, so they are indistinguishable. Therefore, if the upper probe state picks up zero phase shift in the measurement, then the output state of photons 1 and 3 will be ${a|H\rangle }_{1}{|H\rangle }_{3}+{d|V\rangle }_{1}{|V\rangle }_{3}$; if the probe state picks up 2θ phase shift, then the output state of photon 1 and 3 will be ${b|H\rangle }_{1}{|V\rangle }_{3}+{c|V\rangle }_{1}{|H\rangle }_{3}$, up to a normalization factor. The same conclusions can be drawn for the lower probe coherent state.

Thus, when the outcomes of two *X*-quadrature measurements are both 0, we denote this result of PCM as “even”, and the corresponding logical Bell states should be $|\Phi_{L}^{\pm }\rangle$; when the outcomes of two *X*-quadrature measurement are both 2θ, we denote this result of PCM as “odd”, and the corresponding logical Bell states should be $|\Psi_{L}^{\pm }\rangle$. All other cases are considered as failure and discarded.

Further, to discriminate $|\Phi_{L}^{+}\rangle$ with $|\Phi_{L}^{-}\rangle$ and $|\Psi_{L}^{+}\rangle$ with $|\Psi_{L}^{-}\rangle$, we perform Hadamard transformations on the photons, rotating the polarization of each photon by 45 degrees counterclockwise. Then, the four logical Bell states evolve as
(7)$$\begin{array}{cc} |\Phi_{L}^{+}\rangle ⟶\frac{1}{2\sqrt{2}} & (|HHHH\rangle -|HHVV\rangle +|HVHV\rangle -|HVVH\rangle \\ & -|VHHV\rangle +|VHVH\rangle -|VVHH\rangle +|VVVV\rangle ), \end{array}$$
(8)$$\begin{array}{cc} |\Phi_{L}^{-}\rangle ⟶\frac{1}{2\sqrt{2}} & (|HVVV\rangle -|VHVV\rangle +|VVHV\rangle -|VVVH\rangle \\ & -|HHHV\rangle +|HHVH\rangle -|HVHH\rangle +|VHHH\rangle ), \end{array}$$
(9)$$\begin{array}{cc} |\Psi_{L}^{+}\rangle ⟶\frac{1}{2\sqrt{2}} & (|HHHH\rangle -|HHVV\rangle -|HVHV\rangle +|HVVH\rangle \\ & +|VHHV\rangle -|VHVH\rangle -|VVHH\rangle +|VVVV\rangle ), \end{array}$$
(10)$$\begin{array}{cc} |\Psi_{L}^{-}\rangle ⟶\frac{1}{2\sqrt{2}} & (|HVVV\rangle -|VHVV\rangle -|VVHV\rangle +|VVVH\rangle \\ & +|HHHV\rangle -|HHVH\rangle -|HVHH\rangle +|VHHH\rangle ). \end{array}$$

We can see that all components of $|\Phi_{L}^{+}\rangle$ and $|\Phi_{L}^{-}\rangle$ are orthogonal to one another. Therefore, after analyzing the polarizations by means of PBSs, $|\Phi_{L}^{+}\rangle$ and $|\Phi_{L}^{-}\rangle$ will result in completely different click events of $D_{1H},D_{1V},D_{2H}$, and $D_{2V}$. The $|\Psi_{L}^{+}\rangle$ and $|\Psi_{L}^{-}\rangle$ can be discriminated in the same manner. Take the first term of Equation (Equation 7) as an example: if Charlie observes “odd” of the PCM result together with two successive coincidence detections of $D_{1H}$ and $D_{2H}$ separated by Δt, then $\Psi_{L}^{+}$ is identified. We summarize the PCM results and click patterns in Table 2.

However, as discussed in Refs. [27,28,29], the *X*-quadrature measurement in the PCM has an intrinsic probability to misidentify the phase shift of the probe state, resulting in erroneous identification of the parity of input states. We denote this error probability as $P_{e}$, and its analytical expression is $P_{e}=\mathrm{erfc}\left(\sqrt{2}\alpha \theta^{2}\right)/2$ according to Refs. [29,33]. For instance, suppose the parity of photons 1 and 3 is even, then the upper *X*-quadrature measurement will give the intended outcome 0 with probability $1-P_{e}$ while giving the wrong outcome 2θ with probability $P_{e}$. The existence of this error probability indicates unavoidable imperfection of our logical Bell state analyzer and may compromise the secure key rate of our protocol. Nevertheless, $P_{e}$ can be effectively suppressed to the order of $10^{-2}$ by applying small cross-Kerr nonlinearities and high probe intensity, as pointed out in Refs. [28,29]. In Section 5, we study the impact of various $P_{e}$ on the secure key rate where we assume the two *X*-quadrature measurements have identical and independent error probability.

## 4. Security Analysis

The logical BSM apparatus introduced in the above section makes our protocol naturally immune to all detector side attacks as the apparatus is located in the third party. Practically, our protocol still faces two tough security problems in real implementations.

One problem is the potential attack utilizing the 2-dimensional complementary space, span{|HH〉,|VV〉}, in the global 4-dimensional space, which we call complementary space attack. We restrict our discussion in the context of collective attacks [2], i.e, in each round the system is attacked identically and independently of the preceding, and we do not consider the coherent attacks that utilize the complementary space. A malicious eavesdropper Eve who applies the complementary space attack can coherently alter the single-photon-pair state from either party that reaches Charlie to the form of $c_{1}|H^{\prime }V_{T}^{\prime }\rangle +c_{2}|V_{T}^{\prime }H^{\prime }\rangle +c_{3}|H^{\prime }H^{\prime }\rangle +c_{4}|V_{T}^{\prime }V_{T}^{\prime }\rangle$, where the complex coefficients $c_{i}$ are controlled by Eve. Such states can survive the postselection perfectly after Charlie’s delay operation for his $H^{\prime }$ mode and may potentially provide Eve some advantages. As countermeasures, the phase scrambling and sampling procedure, first introduced in Ref. [34], are employed in our protocol.

The phase scrambling procedure is designed to perform projections into the DFS. It applies a Pockel cell driven by quantum random number generators [35,36] to add a phase shift uniformly chosen from $\left\{0,\frac{\pi }{2},\pi ,\frac{3\pi }{2}\right\}$ to Charlie’s $V^{\prime }$ mode. This step can be done with Charlie’s delay operations simultaneously, as illustrated in Figure 1b. If we denote the density operator of the above state $c_{1}|H^{\prime }V_{T}^{\prime }\rangle +c_{2}|V_{T}^{\prime }H^{\prime }\rangle +c_{3}|H^{\prime }H^{\prime }\rangle +c_{4}|V_{T}^{\prime }V_{T}^{\prime }\rangle$ as ρ, then after phase scrambling it becomes
(11)$$\begin{array}{cc} \rho^{\prime } & =\frac{1}{4}\left(U_{0}\rho U_{0}^{†}+U_{\frac{\pi }{2}}\rho U_{\frac{\pi }{2}}^{†}+U_{\pi }\rho U_{\pi }^{†}+U_{\frac{3\pi }{2}}\rho U_{\frac{3\pi }{2}}^{†}\right)\\ \; & =\left(\begin{array}{cccc} {\left|c_{3}\right|}^{2} & 0 & 0 & 0\\ 0 & {\left|c_{1}\right|}^{2} & c_{1}c_{2}^{*} & 0\\ 0 & c_{1}^{*}c_{2} & {\left|c_{2}\right|}^{2} & 0\\ 0 & 0 & 0 & {\left|c_{4}\right|}^{2} \end{array}\right), \end{array}$$
where $U_{\phi }={\left(\begin{array}{cc} 1 & 0\\ 0 & e^{i\phi } \end{array}\right)}^{⊗2},\phi \in \left\{0,\frac{\pi }{2},\pi ,\frac{3\pi }{2}\right\}$ denotes the phase shift operation. As one can see, the phase scrambling maintains the state’s quantum information in the DFS while the coherence with the complementary space is destroyed; the interplay between the two subspaces are thus broken. In other words, the state is projected inside the DFS with some probability. Denote this projection probability for Alice and Bob as $p_{a}$ and $p_{b}$, respectively. As for the events where the states of either party are projected outside the subspace, Alice and Bob assume that Eve has complete knowledge about the key information of these events [34]. Therefore, only events for which both parties’ states remain in the DFS are considered in the key rate formula (see Equation (Equation 16)). Note that the projection results are unknowable in principle using the phase scrambling method. Notwithstanding, as privacy amplification is applied to all effective events where logical Bell states are successfully identified, Alice and Bob do not need to exactly know the projection results.

For the sampling procedure aiming to estimate $p_{a}$ and $p_{b}$, because the polarization controllers act as nothing, Charlie’s measurement is just projections in $\left\{|H^{\prime }\rangle ,|V^{\prime }\rangle \right\}$ basis for each photon. Therefore, employing the intensity setting $\left(\lambda_{a},\lambda_{b}\right)=\left(\mu ,0\right)$ and $\left(0,\mu \right)$, i.e., either party sends vacuum, enables Alice and Bob to effectively measure $p_{a}$ and $p_{b}$ through observing the detectors’ responses in the sampling rounds, respectively. If Eve indeed performs the complementary space attack, Alice (Bob) would find unexpectedly high detection rates for specific click patterns such as $D_{1H(2H)}$ or $D_{1V(2V)}$ clicks twice with time interval Δt in the sampling rounds. We remark that the proportion of the sampling rounds among all rounds can be rather small in the asymptotic scenario. In Appendix C, we present a simple method to calculate $p_{a}$ and $p_{b}$ in a certain scenario. In a realistic experiment, imperfections of optical systems can also give rise to the estimated probabilities $p_{a},p_{b}$ being less than 1 and their dependence on the communication distance [34].

The other problem stems from the probabilistic nature of PDC sources. Recall that the output state of Alice’s type-II PDC source can be written as [37,38]
(12)$$\begin{array}{cc} |\Psi_{a}\rangle & =\frac{1}{\cosh^{2}\chi_{a}}\sum_{n_{a}=0}^{\infty }e^{in_{a}\theta }\sqrt{n_{a}+1}\tanh^{n_{a}}\chi_{a}|\Phi_{n_{a}}\rangle ,\\ |\Phi_{n_{a}}\rangle & =\frac{1}{n_{a}!\sqrt{n_{a}+1}}{\left(H_{1}^{†}V_{2}^{†}+e^{i(\pi K_{a}+\frac{\pi }{2}B_{a})}V_{1}^{†}H_{2}^{†}\right)}^{n_{a}}|\mathrm{vacuum}\rangle \\ \; & =\frac{1}{\sqrt{n_{a}+1}}\sum_{m_{a}=0}^{n_{a}}\exp\left[im_{a}\left(\pi K_{a}+\frac{\pi }{2}B_{a}\right)\right]\\ \; & \;\cdot {|H\rangle }_{1}^{⊗n_{a}-m_{a}}{|V\rangle }_{1}^{⊗m_{a}}{|H\rangle }_{2}^{⊗m_{a}}{|V\rangle }_{2}^{⊗n_{a}-m_{a}}, \end{array}$$
where $\chi_{a}$ is a real number corresponding to Alice’s pumping intensity, $H_{1}^{†}$ denotes the creation operator of the horizontal polarized photon in mode 1, and similar notation for the others. One can calculate that $\lambda_{a}=\sinh^{2}\chi_{a}$ is half of the expected photon pair number of $|\Psi_{a}\rangle$. The single-photon-pair state is of the form $\frac{1}{\sqrt{2}}\left({|HV\rangle }_{12}+\exp\left[i(\pi K_{a}+\frac{\pi }{2}B_{a})\right]{|VH\rangle }_{12}\right)$ and is maximally entangled. However, the multiple-photon-pair states, $|\Phi_{n_{a}}\rangle$ for $n_{a}\geq 2$, open a window for Eve to perform a photon number splitting attack [39,40,41,42], granting her full knowledge about the parties’ key information. Fortunately, this security loophole can be fixed up by decoy-state methods [30,31,32]. Yin et al. proposed a decoy-state method for the original scheme of Boileau et al. and improved the key rate scaling from $O(\eta^{4})$ to $O(\eta^{2})$ [42]. Yet in the MDI scenario, widely-used MDI-decoy-state methods are specifically designed for coherent light sources [19,24,25]; similar analysis for PDC sources is still missing. In this work, we present a three-intensity MDI-decoy-state method with PDC sources to counter the photon number splitting attack.

Here we assume Alice and Bob’s quantum channels are symmetric, i.e., have identical transmittance; the analysis for asymmetric situations can be done with the methods in Refs. [19,43]. As described in Section 2, Alice (Bob) randomly selects $\lambda_{a(b)}$ from {μ,ν,0}, where μ corresponds to signal state and the others correspond to decoy states. After randomizing the phase θ, the output density operator of Alice’s PDC source is
(13)$$\rho_{\lambda_{a}}=\frac{1}{2\pi }\int_{0}^{2\pi }|\Psi_{a}\rangle \langle \Psi_{a}|d\theta =P_{n_{a}}\left(\lambda_{a}\right)|\Phi_{n_{a}}\rangle \langle \Phi_{n_{a}}|,$$
where $P_{n_{a}}\left(\lambda_{a}\right)=\frac{\left(n_{a}+1\right)\lambda_{a}^{n_{a}}}{{\left(1+\lambda_{a}\right)}^{n_{a}+2}}$ is the probability to get $n_{a}$-photon pair. That is, the phase randomized PDC source emits nothing but a classical mixture of $n_{a}$-photon-pair states $|\Phi_{n_{a}}\rangle$, analogous to the fact that a phase randomized coherent state is equivalent to a mixture of Fock states. Therefore, by similar arguments in Refs. [30,31,32], Eve cannot differentiate the original intensity of any signal but only has access to the photon pair number $n_{a}$ and $n_{b}$. Define yield $Y_{n_{a}n_{b}}^{Z(X)}$ to be the conditional probability of an effective event in Z(X) basis given that Alice’s PDC source emits an $n_{a}$-photon-pair state and Bob’s emits an $n_{b}$-photon-pair state, and $e_{n_{a}n_{b}}^{Z(X)}$ to be the corresponding quantum bit error rate (QBER) of such signals. Then, these quantities must have no dependence on the intensity setting $\left(\lambda_{a},\lambda_{b}\right)$, enabling us to write down the following sets of equations: (14)$$\begin{array}{cc} Q_{\lambda_{a}\lambda_{b}}^{Z(X)} & =\sum_{n_{a},n_{b}=0}^{\infty }P_{n_{a}}\left(\lambda_{a}\right)P_{n_{b}}\left(\lambda_{b}\right)Y_{n_{a}n_{b}}^{Z(X)}, \end{array}$$(15)$$\begin{array}{cc} E_{\lambda_{a}\lambda_{b}}^{Z(X)}Q_{\lambda_{a}\lambda_{b}}^{Z(X)} & =\sum_{n_{a},n_{b}=0}^{\infty }P_{n_{a}}\left(\lambda_{a}\right)P_{n_{b}}\left(\lambda_{b}\right)e_{n_{a}n_{b}}^{Z(X)}Y_{n_{a}n_{b}}^{Z(X)}, \end{array}$$
where $Q_{\lambda_{a}\lambda_{b}}^{Z(X)}$ and $E_{\lambda_{a}\lambda_{b}}^{Z(X)}Q_{\lambda_{a}\lambda_{b}}^{Z(X)}$ are overall gain and overall QBER in Z(X) basis given intensity setting $\left(\lambda_{a},\lambda_{b}\right)$, respectively. These quantities are directly accessible in experiments after the sifting step. With the above equations, we obtain the following bounds for the *Z*-basis single-photon-pair yield $Y_{11}^{Z}$ and *X*-basis single-photon-pair QBER $e_{11}^{X}$ following the Gaussian eliminations in Refs. [19,24,25]:$$\begin{array}{cc} {\underset{\bar{}}{Y}}_{11}^{Z}= & \left\{{\left(\frac{\mu }{1+\mu }\right)}^{3}\left[{\left(1+\nu \right)}^{4}Q_{\nu \nu }^{Z}+Q_{00}^{Z}-{\left(1+\nu \right)}^{2}\left(Q_{\nu 0}^{Z}+Q_{0\nu }^{Z}\right)\right]\right.\\ & \left.-{\left(\frac{\nu }{1+\nu }\right)}^{3}\left[{\left(1+\mu \right)}^{4}Q_{\mu \mu }^{Z}+Q_{00}^{Z}-{\left(1+\mu \right)}^{2}\left(Q_{\mu 0}^{Z}+Q_{0\mu }^{Z}\right)\right]\right\}/4{\left(\frac{\mu }{1+\mu }\right)}^{2}{\left(\frac{\nu }{1+\nu }\right)}^{2}\left(\frac{\mu }{1+\mu }-\frac{\nu }{1+\nu }\right),\\ {\bar{e}}_{11}^{X}= & \left[{\left(1+\nu \right)}^{4}E_{\nu \nu }^{X}Q_{\nu \nu }^{X}+E_{00}^{X}Q_{00}^{X}-{\left(1+\nu \right)}^{2}\left(E_{\nu 0}^{X}Q_{\nu 0}^{X}+E_{0\nu }^{X}Q_{0\nu }^{X}\right)\right]{\left(\frac{\mu }{1+\mu }\right)}^{2}\left(\frac{\mu }{1+\mu }-\frac{\nu }{1+\nu }\right)\\ & /\left\{{\left(\frac{\mu }{1+\mu }\right)}^{3}\left[{\left(1+\nu \right)}^{4}Q_{\nu \nu }^{X}+Q_{00}^{X}-{\left(1+\nu \right)}^{2}\left(Q_{\nu 0}^{X}+Q_{0\nu }^{X}\right)\right]\right.\left.-{\left(\frac{\nu }{1+\nu }\right)}^{3}\left[{\left(1+\mu \right)}^{4}Q_{\mu \mu }^{X}+Q_{00}^{X}-{\left(1+\mu \right)}^{2}\left(Q_{\mu 0}^{X}+Q_{0\mu }^{X}\right)\right]\right\}. \end{array}$$

The detailed derivation of the above equations are given in Appendix A. Note that the complementary space attack we considered before clearly does not compromise the fact that $Y_{n_{a}n_{b}}^{Z(X)}$ and $e_{n_{a}n_{b}}^{Z(X)}$ are independent of $\left(\lambda_{a},\lambda_{b}\right)$ and it does not couple with the photon number splitting attack, as the former mainly targets single-photon-pair states, whereas the latter steals information from multiple-photon-pair states. Indeed, Eve can perform complementary space attack to multiple-photon-pair states in principle, but such an attack does not increase her knowledge about the keys thanks to decoy-state methods.

Consequently, equipped with the logical Bell state analyzer and methods to perform projection into the DFS and MDI-decoy-state analysis, we can prove that the security of the protocol is identical to the standard MDI-QKD protocol [10] by using the virtual qubit idea in Lo and Chau’s security proof [44] together with arguments in Ref. [34]. In the asymptotic limit, the key rate formula is given by the GLLP method [45]
(16)$$\begin{array}{cc} R & \geq P_{11}^{Z}{\underset{\bar{}}{Y}}_{11}^{Z}\left[1-h\left({\bar{e}}_{11,AB}^{X}\right)-\left(1-p_{a}p_{b}\right)\right]\\ \; & \;-Q_{\mu \mu }^{Z}f\left(E_{\mu \mu }^{Z}\right)h\left(E_{\mu \mu }^{Z}\right), \end{array}$$
where $1-p_{a}p_{b}$ represents projection loss, ${\underset{\bar{}}{Y}}_{11}^{Z}$ is the lower bound of *Z*-basis single-photon-pair yield, $h\left(x\right)=-x\log_{2}x-\left(1-x\right)\log_{2}\left(1-x\right)$ is the binary Shannon entropy function, *f* > 1 is the error correction inefficiency function, $P_{11}^{Z}=p_{z}^{2}P_{1}{\left(\mu \right)}^{2}$ denotes the joint probability that both Alice and Bob send single-photon-pair states in *Z*-basis with $p_{z}$ being the *Z*-basis probability, and ${\bar{e}}_{11,AB}^{X}$ is the upper bound of *X*-basis single-photon-pair QBER over the events where Alice and Bob’s single-photon-pair states both remain in the DFS. The ${\bar{e}}_{11,AB}^{X}$ can be estimated asymptotically from the following identity (the subscript $\bar{A}$ ($\bar{B}$) denotes projection outside Alice’s (Bob’s) DFS):(17)$$\begin{array}{cc} e_{11}^{X} & =p_{a}p_{b}e_{11,AB}^{X}+\left(1-p_{a}\right)\left(1-p_{b}\right)e_{11,\bar{A}\bar{B}}^{X}\\ \; & +\left(1-p_{a}\right)p_{b}e_{11,\bar{A}B}^{X}+p_{a}\left(1-p_{b}\right)e_{11,A\bar{B}}^{X}. \end{array}$$
It is easy to see that $e_{11,\bar{A}B}^{X}$ and $e_{11,A\bar{B}}^{X}$ are on the order of dark count rate $p_{d}$, whereas $e_{11,\bar{A}\bar{B}}^{X}$ is on the order of $p_{d}^{2}$. Therefore, one can set the upper bound $e_{11,AB}^{X}$ by
(18)$$e_{11,AB}^{X}\leq \frac{e_{11}^{X}}{p_{a}p_{b}}\leq \frac{{\bar{e}}_{11}^{X}}{p_{a}p_{b}}\equiv {\bar{e}}_{11,AB}^{X}.$$
Using Equations (Equation 16) and (Equation 18), we have performed a numerical simulation to analyze the performance of our protocol under various parameter regimes.

## 5. Simulation

The goal of the simulation is to demonstrate the feasibility of the logical Bell state analyzer and MDI-decoy-state method instead of the precise relation between the key rate and fiber length. For this purpose, we assume the absence of birefringence effects and polarization randomization, together with perfect alignment of Alice’s, Bob’s, and Charlie’s reference frames for simplicity, otherwise the number of optical modes would double and the complexity would significantly increase. The robustness of the scheme of Boileau et al. has already be verified experimentally [34], and we believe it is directly applicable in the MDI implementation of our protocol.

In the simulation, we consider inefficient threshold detectors with non-zero dark count rate. Moreover, we neglect the dead time of the detectors and assume that all the detectors are identical for simulation purposes. The cost of sampling is omitted in our simulation, and we also apply a truncation on photon-pair numbers for convenience, with more details given in Appendix B. The simulation parameters are chosen as follows: the detector efficiency $\eta_{d}$ is 50%, the dark count rate $p_{d}$ is $10^{-6}$, the error correction efficiency f=1.1, the attenuation coefficient of the fiber is α=0.16dB/km, the probability of choosing *Z* basis is 15/16, and the intensity probabilities are $p_{0}=2^{-8},p_{\nu }=2^{-7},p_{\mu }=1-p_{0}-p_{\nu }$. The resulting lower bound on key rates versus communication distance between the communication parties with different projection probability $p_{a(b)}$ are illustrated in Figure 3. We have also plotted the key rate of the scheme of Yin et al. [42] with a dashed line for comparison, where the same assumptions are made except for no projection loss. The emission intensities are optimized to be μ=0.12,ν=0.01 in both versions.

From Figure 3, one can see that our MDI version protocol can tolerate higher optical loss than the BB84 version of Yin et al. The maximal distance and key rate decrease as the projection probabilities $p_{a(b)}$ get lower, i.e., the case that Eve performs a stronger complementary space attack. Our protocol can still achieve a decent communication distance even at $p_{a(b)}=0.7$. For $p_{a(b)}$ less than this value, the communication distance and key rate drop rapidly.

Furthermore, we investigate the impact of the error probability $P_{e}$ of *X*-quadrature measurements on the performance with identical parameters as in Figure 3. We range $P_{e}$ from 0 to 0.2 and plot the asymptotic key rates given no projection loss in Figure 4, and find that the communication distance and key rate are hardly influenced when $P_{e}$ is suppressed below $10^{-2}$ as noted before, indicating the robustness of our protocol against this intrinsic imperfection of the logical Bell state analyzer.

## 6. Discussion

Remarkably, one of the key advantages of our protocol is the immunity of all possible detection attacks naturally inherited from the MDI framework. Moreover, the need for nontrivial alignment of reference frames are removed in implementation and the protocol can tolerate the noise induced by birefringence effects in single mode fibers, making it a very promising candidate in intracity quantum networks with accessible fiber infrastructure and space-to-ground quantum communication scenarios [46,47].

From a practical perspective, however, the construction of the logical Bell state analyzer remains a technological challenge because it is generally hard to make large Kerr nonlinearities. Previously we have specified the error probability of *X*-quadrature measurements $P_{e}=\mathrm{erfc}\left(\sqrt{2}\alpha \theta^{2}\right)/2$, which originates from the overlap between the probability distributions of measurement outcomes. Efficient discrimination of logical Bell states only demands small error probability, thus operation in the regime of small cross-Kerr nonlinearities is possible. In view of the QKD system performance, as long as $\alpha \theta^{2}>1.2$, there is little impact on the key rate and communication distance. For realistic coherent states with $\alpha ∼10^{6}$, it is sufficient to apply small nonlinearities on the order of $\theta ∼10^{-3}$ [27,29]. Such size is much smaller than that required to perform controlled-phase gate directly between photons, i.e., θ∼π, and is experimentally achievable via electromagnetically induced transparencies [48,49] or doped optical fibers [50,51].

Another inevitable problem that is of concern to experimentalists is the requirement of four-fold coincidence detection, which results in the poor key rate scaling $O(\eta^{2})$ and short communication distances. We remark that there already exists protocols using hybrid logical basis [52,53], i.e., encoding a logical qubit with two degrees of freedom of a single photon, and their MDI versions have been proposed [54,55]. Although these schemes can only tolerate orthogonal rotations such as reference frame misalignment and cannot overcome the birefringence problem where the phase of polarization modes may shift, they indeed raise the key rate scaling to O(η) and solve the inefficiency problem.

Nevertheless, the main contribution of this work is the first design of MDI-QKD protocol based on the scheme of Boileau et al. [21] by devising a suitable logical Bell state analyzer together with the development of MDI-decoy-state method for PDC sources. Application of heralded PDC sources [56,57] in the setup can increase the efficiency of our systems, although the MDI-decoy-state method needs to be correspondingly revised. We note that one plausible improvement is the mode-pairing scheme [58,59] as in our protocol the two photons are separated in two predetermined time-bins. Pairing two-fold coincidence detections in nonadjacent time-bins can make the key rate scales linearly with transmittance, in principle. What is more, it is intriguing to study whether the error probability in our logical Bell state analyzer will open a security loophole. We leave these as open questions.

## 7. Conclusions

In summary, we have presented a measurement-device-independent quantum key distribution protocol based on decoherence-free subspace with decoy-state method for parametric down-conversion sources and shown its unconditional security. Two major attacks are tackled in the security analysis. A complete logical Bell state analyzer is designed tailored for the decoherence-free subspace. The protocol enables the communication parties to estimate the projection probability into decoherence-free subspace easily with some intensity settings not used for key generation. Through numerical simulations, we show that our scheme can double the secure transmission distance compared with the BB84 version. We also evaluate the effect of inherent imperfections in the logical Bell state analyzer and show that it have negligible influence on the key rate in the asymptotic case.

## Figures and Tables

**Figure 1 entropy-25-00869-f001:**
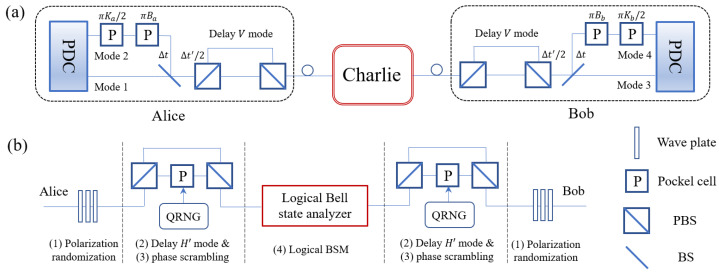
(**a**) Schematic diagram of designed MDI-QKD protocol based on DFS. Alice and Bob each pump their type-II PDC sources with different intensities to generate decoy states. They encode their key bits $K_{a,b}$ and basis bits $B_{a,b}$ using two Pockel cells, respectively, then combine the output modes by means of a beam splitter (BS). After delaying their vertical polarization mode, the signals are sent to Charlie. (**b**) The setup of Charlie’s measurement site. QRNG: quantum random number generator, PBS: polarizing beam splitter.

**Figure 2 entropy-25-00869-f002:**
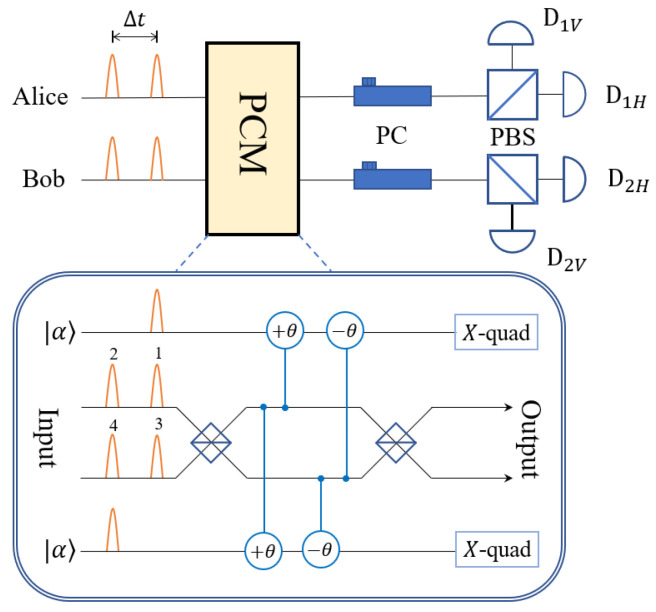
The logical Bell state measurement apparatus; *X*-quad: homodyne measurement of the *X* quadrature; PC: polarization controller that rotates the polarization by 45 degrees such that it acts as a half-wave plate.

**Figure 3 entropy-25-00869-f003:**
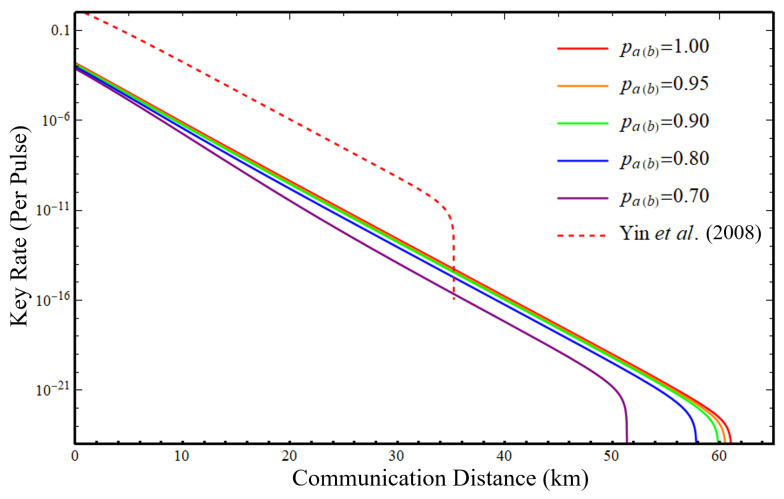
Lower bound on key rate given by Equation (Equation 16) in logarithmic scale for different projection probabilities (solid lines). The key rate of BB84 version protocol using the three-intensity decoy-state method [42] is plotted with a dashed line. The red lines indicate no projection loss.

**Figure 4 entropy-25-00869-f004:**
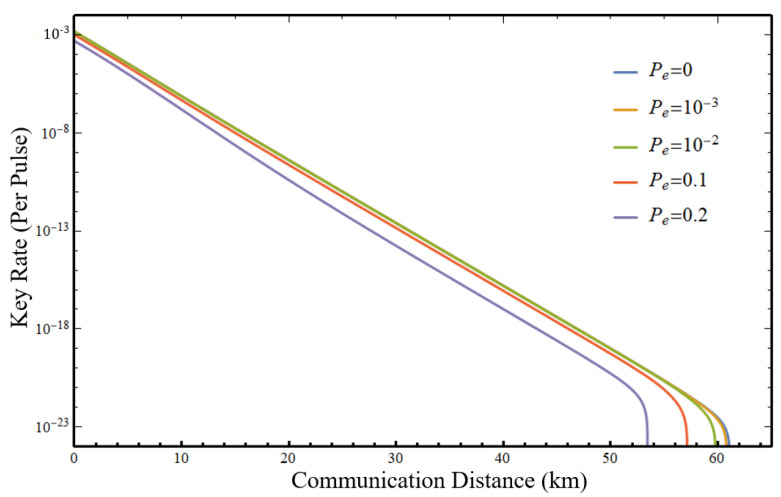
Simulation with various error probability $P_{e}$.

**Table 1 entropy-25-00869-t001:** Either Alice or Bob needs to flip her or his bits based on the successful logical BSM results.

Alice and Bob’s Basis	Logical Bell State			
	$\Phi_{L}^{+}$	$\Phi_{L}^{-}$	$\Psi_{L}^{+}$	$\Psi_{L}^{-}$
*Z* basis	No flip	flip	No flip	flip
*X* basis	flip	No flip	No flip	flip

**Table 2 entropy-25-00869-t002:** The PCM result and click patterns of all four logical Bell states, each of which can result in eight corresponding detection events. The logical Bell states are successfully identified only when a two-fold coincidence detection happens at some time, say $t_{0}$, and another two-fold coincidence happens at time $t_{0}+\Delta t$ subsequently.

PCM Result	$t_{0}$	$t_{0}+\Delta t$	Bell State	PCM Result	$t_{0}$	$t_{0}+\Delta t$	Bell State
even/odd	$D_{1H}D_{2H}$	$D_{1H}D_{2H}$	$\Phi_{L}^{+}$/$\Psi_{L}^{+}$	even/odd	$D_{1H}D_{2H}$	$D_{1H}D_{2V}$	$\Phi_{L}^{-}$/$\Psi_{L}^{-}$
	$D_{1H}D_{2V}$	$D_{1H}D_{2V}$			$D_{1H}D_{2V}$	$D_{1H}D_{2H}$	
	$D_{1H}D_{2H}$	$D_{1V}D_{2V}$			$D_{1H}D_{2H}$	$D_{1V}D_{2H}$	
	$D_{1H}D_{2V}$	$D_{1V}D_{2H}$			$D_{1H}D_{2V}$	$D_{1V}D_{2V}$	
	$D_{1V}D_{2H}$	$D_{1H}D_{2V}$			$D_{1V}D_{2H}$	$D_{1H}D_{2H}$	
	$D_{1V}D_{2V}$	$D_{1H}D_{2H}$			$D_{1V}D_{2V}$	$D_{1H}D_{2V}$	
	$D_{1V}D_{2H}$	$D_{1V}D_{2H}$			$D_{1V}D_{2H}$	$D_{1V}D_{2V}$	
	$D_{1V}D_{2V}$	$D_{1V}D_{2V}$			$D_{1V}D_{2V}$	$D_{1V}D_{2H}$	

## Data Availability

Not applicable.

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
