# Peer review of "Measurement-Device-Independent Quantum Key Distribution Based on Decoherence-Free Subspaces with Logical Bell State Analyzer"

_entropy, 2023, doi:10.3390/e25060869_

Round 1
Reviewer 1 Report
The manuscript is about measurement-device-independent quantum key distribution based on decoherence-free subspaces with a logical Bell state analyzer. In terms of analysis of the proposed method novelty and the results are effectively demonstrated. Further improvements are required. But the presentation must be totally proofread before it is resubmitted. In its present form, there are many grammatical problems and confused wording.
1. In the abstract section, it is suggested to put the numerical simulation analysis explanation.
2. In the introduction section, the last paragraph categorized the section details.
3. Figures and tables captions are too long. It would be better to minimize the caption and explain it in the text.
4. There is a spelling mistake in Form practical perspective, an inevitable problem of our protocol is the requirement of four-fold coincidence detection.
5. Signify the advantage of the proposed scheme and the main contributions of the manuscript in different sections.
6. We develop a logical Bell state analyzer and MDI-decoy-state method [19, 24, and 25] specifically for practical parametric down-conversion (PDC) sources. The author should explain in the comment section how the author's scheme differs from the mentioned reference paper.
7. Usually in the conclusion section it is not preferred to put the references. It is suggested to put the discussion in the result section analysis.
N/A
Reviewer 2 Report
See attached file

Quality of english is fine.
